# Buckling Analysis and Structure Improvement for the Afterburner Cylinder of an Aero-Engine

**Xiaoxia Zheng** [1,2,*], **Yu Zou** [1], **Bohan He** [1], **Jixin Xiang** [1], **Zhiqiang Li** [1,2,3] **and Qiao Yang** [1]

1   College of Aeronautics and Astronautics, Taiyuan University of Technology, Taiyuan 030024, China
2   Institute of Applied Mechanics, College of Mechanical and Vehicle Engineering,
    Taiyuan University of Technology, Taiyuan 030024, China
3   Shanxi Key Laboratory of Material Strength and Structural Impact, Taiyuan University of Technology,
    Taiyuan 030024, China
*   Correspondence: zhengxiaoxia@tyut.edu.cn

**Abstract:** The buckling failure of the afterburner cylinder is a serious safety concern for aero-engines. To tackle this issue, the buckling simulation analysis of the afterburner cylinder was carried out by using finite element method (FEM) software to obtain the buckling mode and critical buckling loads. It was found that the afterburner cylinder was susceptible to buckling when subjected to differential pressure or the compressive force of the rear flange. Buckling would occur when the differential pressure reached 0.4 times the atmospheric pressure or when the axial compressive force on the rear flange reached 222.8 kN. Buckling was also found at the front of the cylinder under the auxiliary mount load. Additionally, under various loads on the rear flange, buckling occurred in the rear section, with the buckling mode being closely related to the load characteristics. Based on the simulation results and structural design requirements, two structural improvements were proposed, including the wall-thickening scheme and the grid reinforcement scheme. FEM simulation analysis results showed that both schemes would improve the rigidity and stability of the afterburner cylinder. For the 0.3 mm increase in the wall thickness scheme, the critical buckling load increased by 17.86% to 66.4%; for the grid reinforcement scheme, the critical buckling load increased by 169% to 619%. Therefore, the grid reinforcement scheme had a stronger anti-buckling ability and was deemed the optimal solution. The findings of this paper could provide technical support for the structural design of large-sized and thin-walled components of aero-engines.

**Keywords:** aero-engine; the afterburner cylinder; buckling mode; critical buckling load; structural improvement

## 1. Introduction

The aero-engine is a highly complicated and precise thermal-rotating machine, often referred to as the "pearl in the crown of modern industry". The technical level of the aero-engine reflects the industrial development and technical evolution of a country. Aside from that, the performance of an aircraft is highly dependent on the aero-engine, since it decides the thrust and the fuel consumption.

The afterburner is n unit body located between the turbine and the nozzle. The main purpose is to achieve additional thrust after the aero-engine reaches the maximum state. To be specific, the exhaust is reheated to a very high temperature in the afterburner to increase the instantaneous thrust and improve the maneuverability and operability. Supersonic combat and some tactical actions would be much easier and more approachable.

The afterburner cylinder is the core component of the afterburner unit featuring the thin-walled and load-bearing component. It not only distributes loads through the auxiliary mount connecting with the airplane, it also includes various loads transferring from nozzle, deflection, and other adjustments. Therefore, the structure of the afterburner cylinder is very complex, with multi-loading conditions. To be specific, the afterburner cylinder works

in a harsh environment and is subject to aerodynamic loads, temperature loads, maneuver loads, and other heavy loads. Considering the structural form and load characteristics of the afterburner cylinder, the global or local buckling is more likely to occur during engine tests and field usage. According to the requirements of aero-engine strength design and structural integrity, buckling analysis should be carried out at the initial stage of the afterburner cylinder design.

Buckling refers to the equilibrium transfer of structures. When the stable equilibrium state is disturbed by any small external force, the structure works in an unbalanced state, which is called structural buckling. Buckling instability is one of the most common failure modes of thin-walled casings for aero-engines. When a thin-walled casing is subjected to loads, such as pressure difference, axial force, torque, and bending moment, the casing will produce buckling deformation instantly if the complex loads reach or exceed the critical loads. Consequently, the casing deflection will increase sharply, which poses a serious threat to the safety of the components.

Buckling is one of the primary failure modes in thin-walled components, frequently leading to severe consequences. The buckling behavior of thin-walled components has been extensively studied by researchers worldwide. Literatures [1–4] investigated the buckling of spherical and hemispherical shells in engines, finding that buckling occurred suddenly upon reaching the critical buckling loads. The buckling performance of the thin-walled beam was studied by researchers [5–10]. Elishakoff provided an overview of the elastic stability of shells [5], while Einafshar et al. [6] proposed an efficient, one-dimensional finite element model for the buckling and post-buckling analysis of thin-walled beams under axial bending moments. Ziane et al. [7] examined the post-buckling behavior of simply supported steel beams with rectangular hollow sections under large torsion and section distortion. Yang et al. [8] utilized the spline finite element method (FEM) to analyze the static and dynamic buckling of thin-walled beams, demonstrating that the method was efficient in assessing structural frames. He et al. [9] employed the FS-TMM to analyze the buckling performance of thin-walled members with open-branched cross-sections.

Thin-walled components are susceptible to buckling under specific loads. Bourihane et al. [10] investigated the buckling of thin-walled beams with open sections subjected to arbitrary loads, while Jiao [11] studied the buckling of cylindrical shells under axial compression loads. Vasilikis [12] examined the buckling of cylindrical shells subjected to external pressure. Composite components are also prone to buckling instability [13–18]. Maali [17] assessed the buckling and post-buckling behaviors of cylindrical shells made from different materials. Shen et al. [18] investigated the buckling and post-buckling behaviors of composite cylindrical shells under hydrostatic pressure, indicating that reinforcement design effectively ensures the load-carrying capacity. Bin Kamarudin et al. [19] studied the buckling of thin-walled composite structures by using the FEM software ABAQUS. The critical buckling load was obtained by using eigenvalue linear buckling, and the results were optimized based on parameters. Zhang et al. [20] developed a local buckling analysis method and studied the buckling analysis of thin-walled metal liners of cylindrical composites. Ehsani [21] set up the numerical model of laminated composites and received the elastic buckling load by using the Ritz method.

The large-sized and thin-walled afterburner cylinder of an aero-engine was taken as the research object, and the buckling instability simulation analysis of the afterburner cylinder under different loads was carried out. The critical buckling load, buckling mode, and displacement contour were given. Based on the buckling simulation results and combined with engineering practical experience, two improvement schemes were given, and the best scheme was obtained through comparative analysis.

The results of this paper can not only guide the design and structural optimization of large-sized and thin-walled components for aero-engines, it can also provide important technical support for military turbofan aero-engines with a high thrust–weight ratio and a small bypass ratio.

## 2. Buckling Theory

The structural stability theory includes the classical linear elastic theory and the nonlinear buckling theory. The classical theory, proposed by Euler and Lagrange, was based on linear elasticity. Nevertheless, it had limitations, and theoretical results differed from experimental results. The nonlinear buckling theory, including nonlinear large deflection, nonlinear pre-buckling consistency, and initial post-buckling theory, was developed to overcome these limitations. For example, Donnell [22] and Karman and Tsien [23] proposed the nonlinear large-deflection theory and studied post-buckling morphology, uncovering the fact that unstable structures had highly unstable nonlinear post-buckling characteristics.

Nowadays, buckling analysis generally includes eigenvalue buckling (linear buckling) and post-buckling (nonlinear buckling). Eigenvalue buckling is based on the linear elastic theory without considering nonlinear behavior or initial structural damage. Research has shown that the critical value of buckling obtained from eigenvalue buckling is larger than the actual experimental value. However, it is still a common and acceptable method to analyze buckling instability, due to its low cost.

For linear (eigenvalue) buckling analysis, the main solving steps are as follows.

Load displacement relationship for the linear elastic solution:

$$\{F_0\} = [K_e]\{u_0\} \tag{1}$$

where $\{F_0\}$ is the load matrix, $\{u_0\}$ is the displacement matrix, and $[K_e]$ is the elastic stiffness matrix.

Form of balance equation based on the increment:

$$\{\Delta F\} = [[K_e] + [K_\sigma(\sigma)]]\{\Delta u\} \tag{2}$$

where $\{\Delta F\}$ is the load increment, $\{\Delta u\}$ is the corresponding displacement increment, and $[K_\sigma(\sigma)]$ is the initial stress matrix $\{\sigma\}$ under the corresponding stress state.

Assuming that the linear buckling behavior is a linear function of the applied load $\{F_0\}$, it can be established that:

$$\{F\} = \lambda\{F_0\}, \{\sigma\} = \lambda\{\sigma_0\}, \{u\} = \lambda\{u_0\} \tag{3}$$

We can derive that:

$$[K_\sigma(\sigma)] = \lambda[K_\sigma(\sigma_0)] \tag{4}$$

The balanced equation can be converted into:

$$\{\Delta F\} = [[K_e] + \lambda[K_\sigma(\sigma_0)]]\{\Delta u\} \tag{5}$$

When the structure is under the critical buckling load, a small external disturbance will lead to large changes in structural deformation:

$$\{\Delta F\} = [[K_e] + \lambda[K_\sigma(\sigma_0)]]\{\Delta u\} = 0 \tag{6}$$

To set up Formula (6), we can use:

$$\Delta[[K_e] + \lambda[K_\sigma(\sigma_0)]] = 0 \tag{7}$$

Then, the $\lambda$ can be obtained; $\lambda$ is the critical buckling eigenvalue.

The critical buckling load of the structure can then be obtained:

$$F_{cr} = \lambda \cdot F \tag{8}$$

If the structure has undergone significant deformation before buckling occurs, the linear buckling theory is no longer applicable, indicating that the post-buckling behavior must be considered. Post-buckling is a nonlinear behavior that requires the consideration of

material nonlinearity, geometric nonlinearity, and initial defects. In finite element analysis, the arc length method is commonly used, and the arc length is adopted as a variable. It is implemented as the length of the static equilibrium path in the load-displacement space to solve the post-buckling problem.

However, the computational cost of post-buckling analysis is higher than that of linear buckling analysis. In order to save computational costs, improve computational efficiency, and ensure structural safety, linear buckling has been used widely in engineering design [24–26]. Not only can it evaluate critical loads quickly, it can also obtain buckling modes and guide structural anti-buckling design. To ensure the safety of components, as well as to save costs, this paper uses the linear buckling method for simulation analysis.

## 3. Buckling Analysis of the Afterburner Cylinder

### 3.1. Structure and Loads

#### 3.1.1. Structure Description

The afterburner cylinder is an annular, thin-walled structure located at the outermost end of the afterburner unit, as shown in Figure 1. It is located between the afterburner diffuser and the tail nozzle. The primary function of the afterburner cylinder is to form internal and external airflow passages and transfer some of the engine loads to the aircraft through the auxiliary mount. To increase local rigidity, a load-bearing ring is usually placed on the afterburner cylinder, and auxiliary mounts are placed on the load-bearing ring to endure big loads. The mounts are welded at the corresponding positions of the afterburner cylinder and connected to the actuating cylinder or pull rod to transfer the loads from the aero-engine to the airplane.

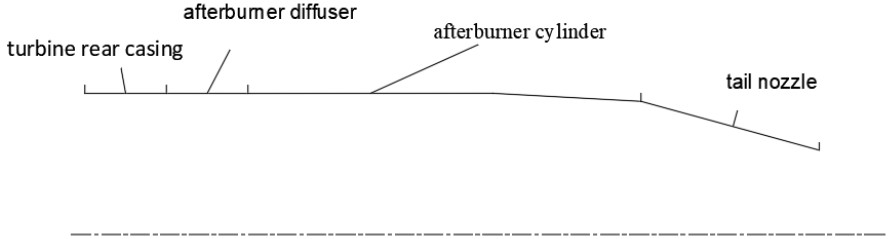

**Figure 1.** The position of the afterburner cylinder.

The structure of the afterburner cylinder is depicted in Figure 2, the green represents the outer wall surface, while the purple represents the inner wall surface. The overall wall thickness of the afterburner cylinder was $t$ = 1.2 mm, and the axial length was $L$ = 1000 mm. The cylinder was divided into two sections, with the front half of the cylinder being cylindrical, while the second half was conical. The radii of the front and second part of the cylinder were 500 mm and 480 mm, respectively, the axial lengths of the front part and second part were 333 mm and 667 mm, respectively, and the material was titanium alloy.

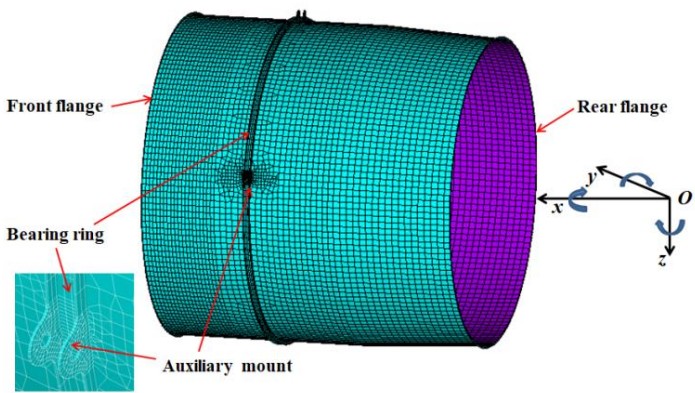

**Figure 2.** The diagram of the afterburner cylinder.

3.1.2. The Loads

Based on the location and the needs of the loads' transfers, the load types of the afterburner cylinder are complex. The main loads of the afterburner cylinder are shown in Table 1.

**Table 1.** The main load types of the afterburner cylinder.

| Load Types | Description |
|---|---|
| Auxiliary mount | One or two mounts are loaded |
| Rear flange | Axial force, vertical force, lateral force, bending moment, torque |
| Differential pressure | External pressure is greater than internal pressure |

From the results of the strength test and the load analysis of the aero-engine, it can be seen that the auxiliary mount load, the axial compressive force on the rear flange, and the differential pressure on the casing wall would lead to buckling for the afterburner cylinder.

The purpose of this paper was to qualitatively characterize the buckling instability of the afterburner cylinder and provide useful improvements, based on engineering experience. The calculated load was a unit load, not the actual load. The load list is presented in Table 2, and the load coordinate system is shown in Figure 2. It can be seen that the $x$-axis was positive along the forward direction of the aircraft, the $z$-axis was positive along the vertical direction, and the $y$-axis was determined by the right-handed coordinate system.

**Table 2.** Loads list of the afterburner cylinder.

| Load Condition | Load Types | | Load Value [1] |
|---|---|---|---|
| 1 | Auxiliary mount loads | One mount is loaded | $F_{z1} = -1000$ N |
| 2 | | Two mounts are loaded | $F_{z1} = -1000$ N, $F_{z3} = -1000$ N |
| 3 | Rear flange loads | Axial compressive force | $Fx = 1000$ N |
| 4 | | Lateral force | $Fy = 1000$ N |
| 5 | | Torque | $Mx = 1000$ N · m |
| 6 | | Bending moment | $My = 1000$ N · m |
| 7 | Differential pressure | | $q = 1$ MPa |

[1] The direction of the force is shown in Figure 2.

According to engineering experience, the temperature distribution of the afterburner cylinder gradually increases along the axial direction. The temperature distribution is depicted in Figure 3. It can be seen that the temperature difference was about 127 K, and the temperature gradient was relatively low. The main reason was that an annular heat shield was installed inside the wall of the afterburner cylinder. As a result, the high temperature was separated from the turbine exhaust. The buckling analysis also took into account the effect of temperature on material properties in this simulation.

*3.2. The Buckling Analysis of the Afterburner Cylinder*

Based on the buckling issue of the afterburner cylinder in the engine test or field usage, the buckling simulation analysis of the afterburner cylinder was carried out under various working conditions. The buckling modes and critical buckling loads were obtained.

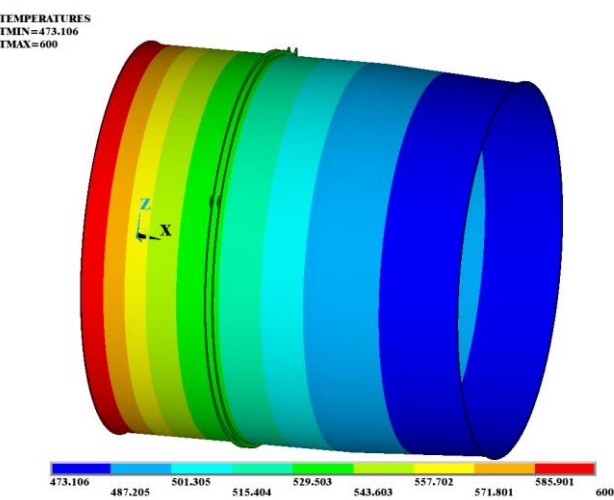

**Figure 3.** The distribution of temperature (K).

### 3.2.1. The Buckling Analysis under Auxiliary Mount Loads

The buckling modes under different load conditions were obtained and are demonstrated in Figures 4 and 5. In the figures, the DMX represented the maximum displacement and SMX represented the specified maximum solution. Under a single auxiliary mount load, the critical buckling load coefficient was $\lambda$ = 31.31, and the critical buckling load was $Fcr$ = 31.31 $\times$ 1000 = 31,310 N = 31.31 kN. Local buckling was found on the thin-walled cylinder directly in front of the mount load, and the buckling occurred in the form of depression and drumming alternately along the oblique direction. The analysis showed that the load of a single auxiliary mount would create a torque load on the front of the cylinder, leading to the local buckling of the front cylinder.

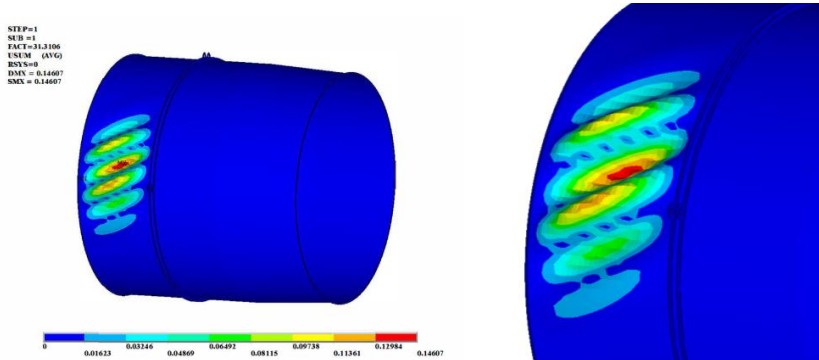

**Figure 4.** The contour of buckling mode under one auxiliary mount load.

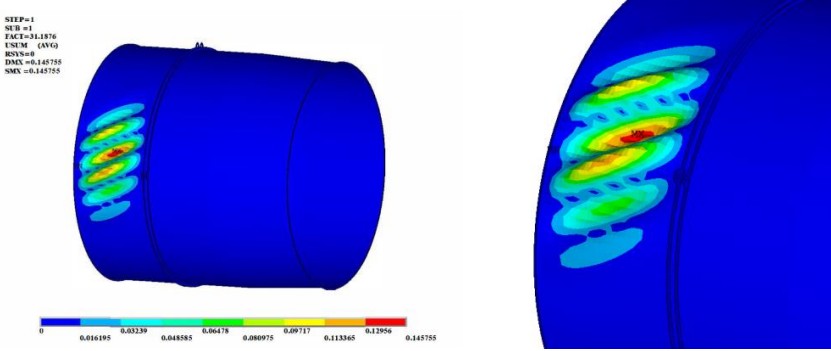

**Figure 5.** The contour of buckling mode under two auxiliary mount loads.

Under the combined loads of the left and right auxiliary mounts, the critical buckling load coefficient was λ = 31.19, and the critical buckling load was *Fcr* = 31.19 × 1000 = 31,190 N = 31.19 kN. Because the loads of the left and right auxiliary mounts were the same, the buckling instability randomly appeared on the front cylinder corresponding to one mount, and the buckling displacements occurred alternately, with depressions and drum-shaped bags along the oblique direction. The buckling modes and critical buckling load were basically similar to those under the load of a single mount. Results showed that when the load was large enough, temporary equilibrium failure and buckling would occur when the mounts on both sides were loaded simultaneously. If the loads of the auxiliary mounts on both sides were different, a torque would be formed on the cylinder, and buckling would occur in the front section of the cylinder corresponding to the mount with the bigger load.

### 3.2.2. Buckling Analysis under Loads of the Rear Flange

The loads on the rear flange of the afterburner cylinder mainly included the axial aerodynamic load caused by thrust, the axial force, lateral force, vertical force, bending moment, and the torque transmitted by aircraft maneuvering flight, and the buckling simulation analysis was carried out under these loads. The calculating loads were taken from load condition 3 to condition 6 from Table 2.

The buckling modes and critical buckling loads of the afterburner cylinder were obtained by using finite element simulation analysis under various loads on the rear flange, as detailed in Table 3 and Figures 6–9.

**Table 3.** Buckling analysis under the loads of the rear flange.

| Load Types | Load Value | Critical Buckling Eigenvalue (λ) | Critical Buckling Loads | Buckling Modes |
|---|---|---|---|---|
| Axial compressive force | $Fx = 1000$ N | 222.8 | 222.8 kN | Buckling in the rear section of the cylinder [2], $m = 6$, $n = 1$ |
| Lateral force | $Fy = 1000$ N | 39.2 | 39.2 kN | The rear part of the cylinder is unstable, and the buckling deformation is related to the load direction. |
| Torque | $Mx = 1000$ N · m | 85.1 | 85.1 kN · m | Buckling in the rear section of the cylinder, $m = 15$, $n = 1$ |
| Bending moment | $My = 1000$ N · m | 59 | 59 kN · m | The rear part of the cylinder is buckling, and the buckling deformation is related to the load direction. |

[2] $m$ is the circumferential wave number; $n$ is the axial half-wave number.

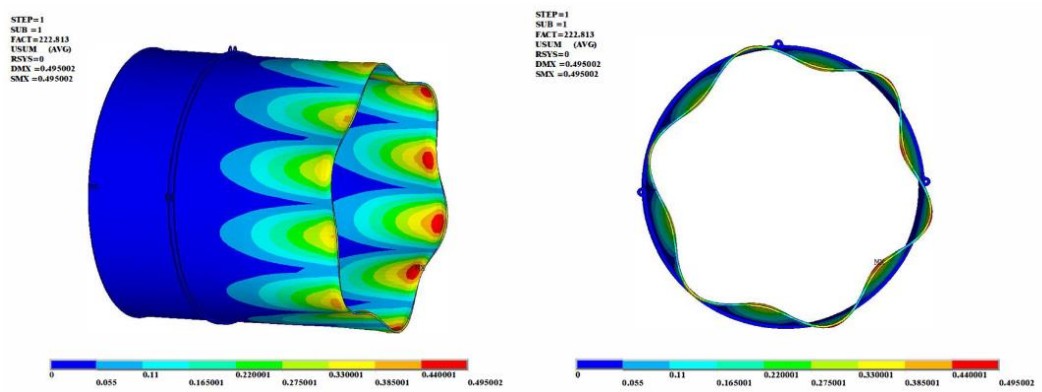

**Figure 6.** The contour of buckling mode under axial compressive force of the rear flange.

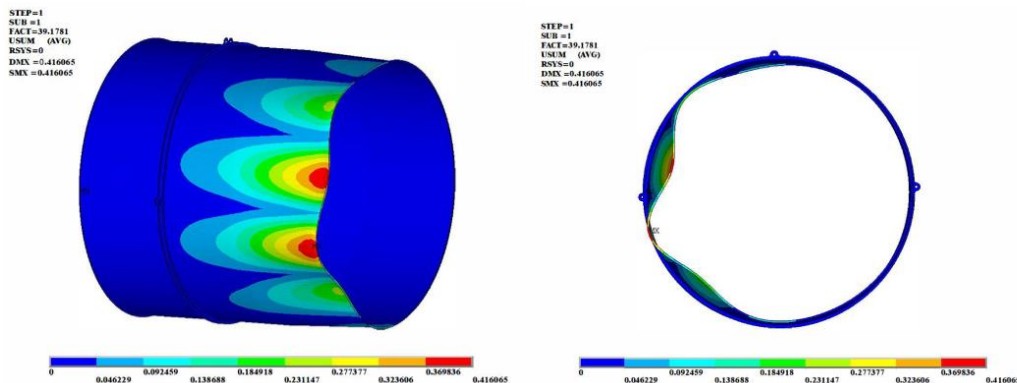

**Figure 7.** The contour of buckling mode under lateral force of the rear flange.

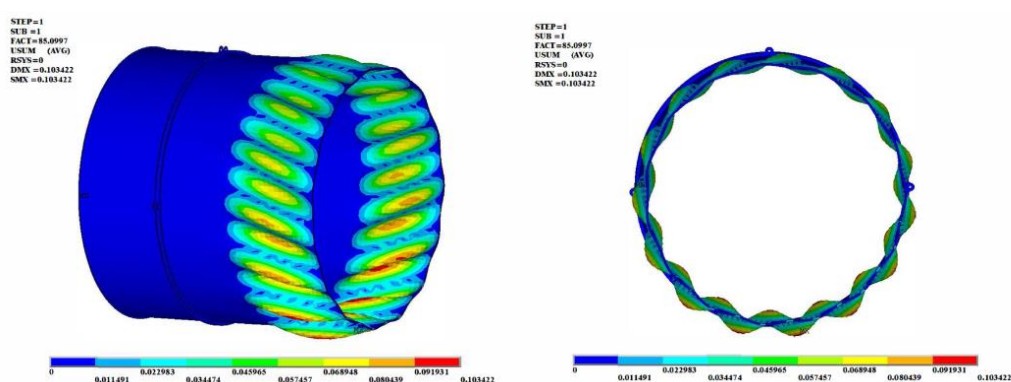

**Figure 8.** The contour of buckling mode under torque of the rear flange.

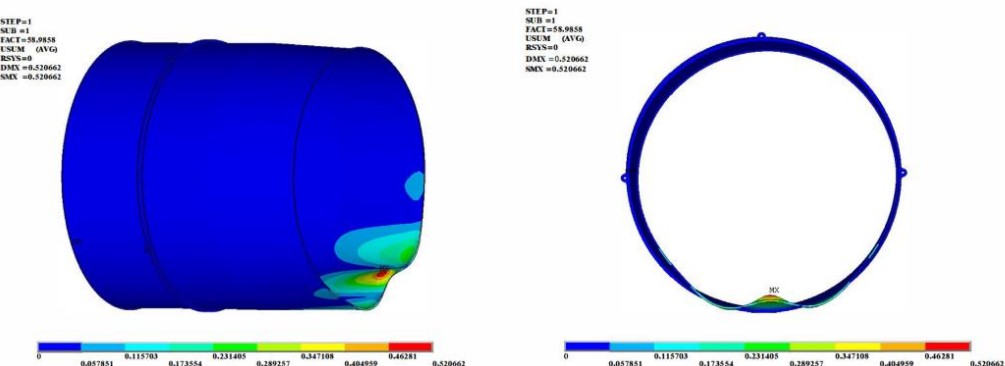

**Figure 9.** The contour of buckling mode under bending moment of the rear flange.

The main results were as follows:

- Under the loads of the axial compressive force or torque, the buckling modes of the afterburner cylinder showed a regular wavy distribution, with an axial half-wave number of $n = 1$. The circumferential wave number was dense under the load of the torque, with a wave number of $m = 15$, but under the axial compressive force, the circumferential wave number was $m = 6$.
- Under the load of the axial compressive force, the critical buckling load of the cylinder was 222.8 kN. The load value was relatively small, compared to the axial load at the flange, indicating that the cylinder was prone to buckling under the axial compressive force of the rear flange.
- Under the load of the lateral force, the critical buckling load was 39.2 kN. This load value was also small, which indicated the cylinder was prone to buckling.

- Buckling only occurred near the rear flange under the load of the bending moment or the lateral force, and the buckling deformation was closely related to the direction of the loads.
- The buckling occurred in the rear section of the cylinder under various loads of the rear flange.

### 3.2.3. Buckling Analysis under Differential Pressure

Based on the fact that the thin-walled cylinder was prone to buckling when external pressure exceeded internal pressure, a simulation of the afterburner cylinder's buckling instability under differential pressure ($q$ = 1 MPa) was carried out. The finite element simulation results indicated that the buckling occurred at the back of the cylinder, with the buckling deformation appearing alternately in the wave form of the drum bag and the depression, as shown in Figure 10. The axial half-wave number was $n$ = 1, and the circumferential wave number was $m$ = 13.

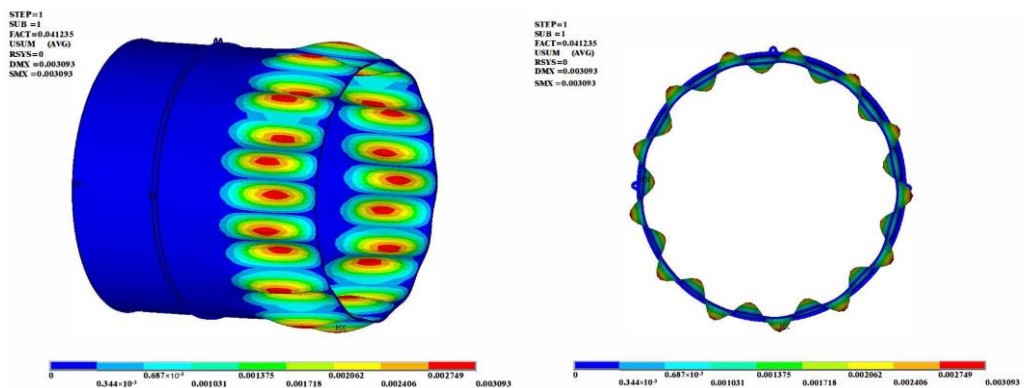

**Figure 10.** The contour of buckling mode under differential pressures (The external pressure was greater than the internal pressure).

The critical buckling load coefficient was $\lambda$ = 0.041235, and the critical buckling load was $q_{cr}$ = 0.041235 MPa, which was relatively small and was equal to 0.4 times the atmospheric pressure. The analysis revealed that, for large-sized and thin-walled cylinders similar to the afterburner cylinder, buckling instability was likely to occur when the external pressure was greater than the internal pressure.

### 3.3. Buckling Analysis and Summary

Based on the analysis of this section, the following conclusions were drawn:

1. Under the load of a single auxiliary mount, local buckling occurred in the front section of the cylinder corresponding to the location of the load, with buckling displacement distributed along the oblique direction. The buckling was caused by the torque generated by the load of the mount in the front section of the cylinder.
2. Under the loads of the auxiliary mounts on both the left and right sides, the casing would still occur buckling. If the loads on both sides were equal, buckling would occur randomly in the front cylinder corresponding to one side of the mount. If the loads on both sides were not equal, buckling would occur in the cylinder corresponding to the larger load of the mount.
3. Under the loads of one or two auxiliary mounts, the critical buckling load of the afterburner cylinder was very close.
4. Under the loads of the rear flange, buckling occurred near the rear flange, and buckling mode was closely related to the load form.
5. Under the loads of axial compressive force or torque on the rear flange, the buckling displacement of the cylinder showed a regular wave distribution, with an axial half-wave number of $n$ = 1. The circumferential wave number under the load of the torque was much bigger than that under the load of the axial compressive force.

6. Under the load of the bending moment or the lateral force on the rear flange, buckling only occurred in the local casing near the rear flange, with buckling deformation closely related to the direction of the load.
7. The buckling of the afterburner cylinder was highly likely to occur under external pressure bigger than the internal pressure or under axial compressive force.

The afterburner cylinder of the aero-engine encountered buckling failure during the overall test and the structural strength tests. The buckling mode and displacement distribution were consistent with the results in this paper. Other thin-walled cylinders of aero-engines (such as the afterburner inner cone) also suffered from buckling failures under those loads, and troubleshooting analysis showed that the differential pressure and the compressive force of the rear flange easily caused the buckling of thin-walled components, which confirmed the rationality of the buckling simulation analysis in this paper.

## 4. Improvement and Discussion

Theoretical analysis indicated that the primary reasons for the buckling of the afterburner cylinder were the thin walls and the weak rigidity. The most effective method to improve the design was to increase the local or overall rigidity. Considering the buckling mode of the afterburner cylinder under various loads, the load characteristics, space limitations, and weight requirements, two structural improvements, including wall thickening and grille reinforcement, were proposed. Buckling simulation analyses were performed to evaluate the effectiveness of these two improvement schemes.

### 4.1. Cylinder Wall-Thickening Scheme

The original wall thickness of the afterburner cylinder was $t = 1.2$ mm. The buckling modes and critical buckling loads were studied by increasing the wall thickness by 0.1 mm, 0.2 mm, and 0.3 mm. Finite element analysis showed that the buckling mode of the afterburner cylinder did not change with the increase in wall thickness. However, the critical buckling loads increased significantly. Tables 4–6 showed the critical buckling load for each case. Figure 11 presented the relationship between increasing wall thickness and critical buckling loads.

**Table 4.** Buckling analysis under the loads of the auxiliary mount/differential pressure.

| Load Types / Wall Thickness | Auxiliary Mount | | Differential Pressure | |
|---|---|---|---|---|
| | $\lambda$ | Increment (%) [3] | $\lambda$ | Increment (%) |
| $t = 1.2$ mm (Original scheme) | 31.31 | — | 0.041235 | — |
| $t = 1.3$ mm | 37.58 | 20 ↑ | 0.0434 | 5.25 ↑ |
| $t = 1.4$ mm | 44.49 | 42.1 ↑ | 0.046 | 11.56 ↑ |
| $t = 1.5$ mm | 52.1 | 66.4 ↑ | 0.049 | 18.83 ↑ |

[3] ↑ represents the increase in $\lambda$.

**Table 5.** Buckling analysis under the loads of the rear flange-1.

| Load Types / Wall Thickness | Axial Compressive Force | | Lateral Force | |
|---|---|---|---|---|
| | $\lambda$ | Increment (%) | $\lambda$ | Increment (%) |
| $t = 1.2$ mm (Original scheme) | 222.8 | — | 39.2 | — |
| $t = 1.3$ mm | 249 | 11.76 ↑ | 41.4 | 5.61 ↑ |
| $t = 1.4$ mm | 276.8 | 24.24 ↑ | 43.7 | 11.48 ↑ |
| $t = 1.5$ mm | 307.4 | 37.97 ↑ | 46.2 | 17.86 ↑ |

↑ represents the increase in $\lambda$.

**Table 6.** Buckling analysis under the loads of the rear flange-2.

| Load Types / Wall Thickness | Torque | | Bending Moment | |
|---|---|---|---|---|
| | λ | Increment (%) | λ | Increment (%) |
| *t* = 1.2 mm (Original scheme) | 85.1 | — | 59 | — |
| *t* = 1.3 mm | 103.4 | 21.50 ↑ | 66.7 | 13.05 ↑ |
| *t* = 1.4 mm | 118.8 | 39.60 ↑ | 75.1 | 27.29 ↑ |
| *t* = 1.5 mm | 135.7 | 59.46 ↑ | 84.1 | 42.54 ↑ |

↑ represents the increase in λ.

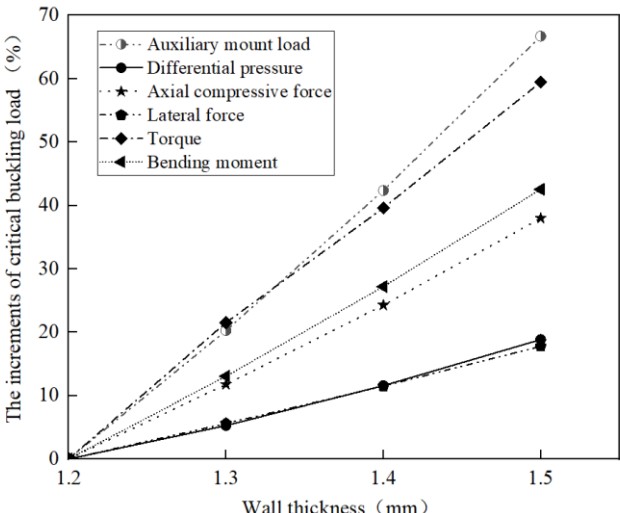

**Figure 11.** The relationship between wall thickness and the increments of critical buckling loads.

Simulation analysis results showed that:

1. The critical buckling load increased linearly as the wall thickness increased under all loads.
2. Under the load of the torque or the auxiliary mount, increasing the wall thickness of the cylinder resulted in a relatively higher increase in the critical buckling load.
3. Under the load of differential pressure or lateral force, increasing the wall thickness of the cylinder resulted in a relatively smaller increase in the critical buckling load.

### 4.2. Grille Reinforcement Scheme

From practical design experience, it was established that incorporating grille reinforcements on the exterior surfaces of thin-walled casings effectively enhances their stiffness, strength, and stability. The grille reinforcements featured dimensions of 5 mm in height and 1.5 mm in width. Figure 12 illustrates the structural design of the grille reinforcement applied to the afterburner cylinder in an aero-engine gas turbine system.

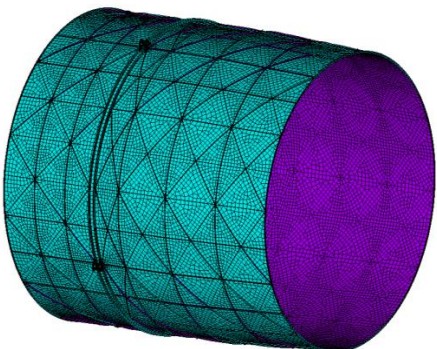

**Figure 12.** The structure of the afterburner cylinder by grille reinforcement.

The finite element buckling simulation analysis was conducted on the cylinder reinforced with grille structures. The applied loads were obtained from Table 2, and the resulting buckling modes and critical buckling loads of the cylinder under various loading conditions were determined. The deformation contours are illustrated in Figure 13. The comparison of the critical buckling loads is presented in Table 7 and Figure 14.

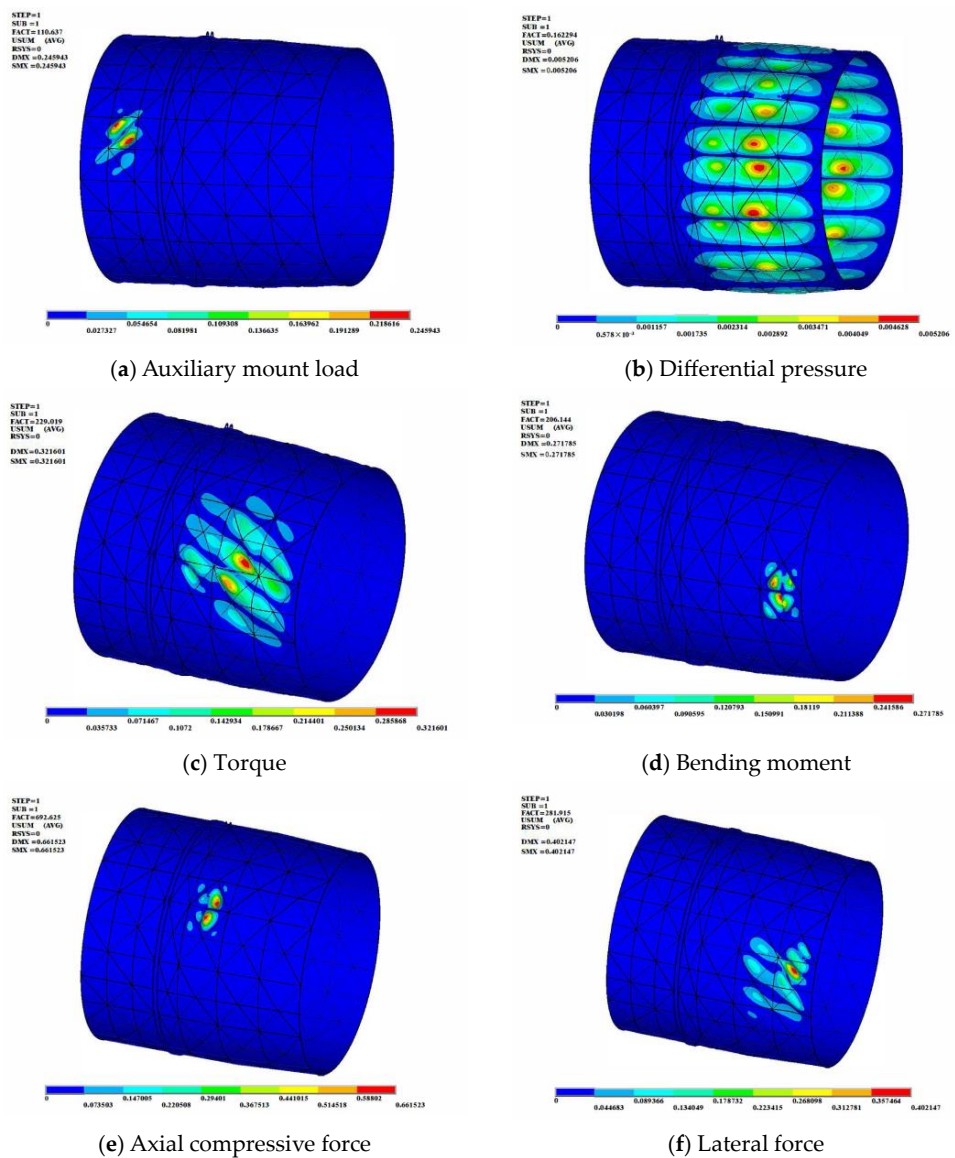

(**a**) Auxiliary mount load

(**b**) Differential pressure

(**c**) Torque

(**d**) Bending moment

(**e**) Axial compressive force

(**f**) Lateral force

**Figure 13.** The contours of buckling mode by grid reinforcement.

**Table 7.** The critical buckling loads by grid reinforcement.

| Load Types | Original Scheme λ | Grille Reinforcement λ | Increment (%) |
|---|---|---|---|
| Auxiliary mount | 31.31 | 110.6 | 253 ↑ |
| Axial compressive force | 222.8 | 692.6 | 211 ↑ |
| Differential pressure | 0.04123 | 0.162 | 293 ↑ |
| Torque | 85.1 | 229.0 | 169 ↑ |
| Bending moment | 59 | 206.1 | 249 ↑ |
| Lateral force | 39.2 | 281.9 | 619 ↑ |

↑ represents the increase in λ.

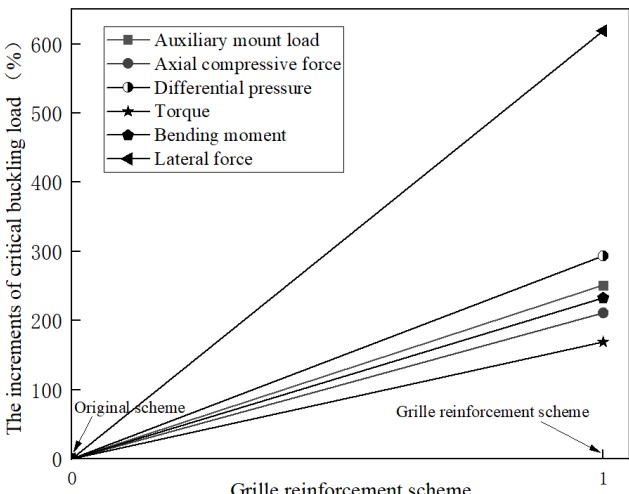

**Figure 14.** The increase in critical buckling load by grid reinforcement.

The main conclusions were as follows:

1. Under the loading conditions of the auxiliary mount or differential pressure, the buckling mode remained largely consistent with the original design, while the critical buckling load increased significantly.
2. Under various rear flange loading conditions, the buckling modes of the reinforced cylinder exhibited considerable deviation from the original design. The critical buckling loads increased by 169% to 619%, signifying a substantial enhancement in the structure's load-bearing capacity.

These findings demonstrate the effectiveness of grille reinforcements in improving the buckling performance of thin-walled casings in aero-engine systems. The increased critical buckling loads contribute to the overall structural stability and reliability of the engine components under various operational conditions.

*4.3. Discussion*

Based on the analysis of the two improvements and the comparative analysis with the original scheme, the conclusions were as follows:

1. Compared with the original scheme, the wall-thickening scheme could not alter the buckling mode of the cylinder. As the wall thickness increased, the critical buckling load under all loading conditions exhibited a linear growth. With a 0.3 mm increase in wall thickness, the critical buckling load increased by 17.86% to 66.4%, but the increase in wall thickness would result in a significant increase in structural weight, which would affect the engine's thrust-to-weight ratio.
2. For the grille reinforcement scheme, the buckling modes under the auxiliary mount load and the differential pressure load remained similar to the original scheme. However, the buckling modes under the rear flange loads exhibited substantial differences, compared to the original design. Under various loading conditions, the critical buckling load of the cylinder increased by 169% to 619%, with a more significant increase in critical buckling load compared to the wall-thickening approach.

Compared with the original scheme, the two improvement schemes proposed in this paper could effectively enhance the overall rigidity and stability of the afterburner cylinder, and the critical buckling load would be effectively increased. The grille reinforcement scheme made a greater increase in the critical buckling load than the wall-thickening scheme, and the wall-thickening scheme would result in a significant increase in quality, which would seriously affect the improvement of thrust-to-weight ratio for the aero-engine.

Taking into account practical engineering experience and the comparative results of the two schemes, the grille reinforcement scheme proved to be more effective than the

wall-thickening scheme in enhancing the structural stability and load-bearing capacity of thin-walled casings in aero-engine systems.

## 5. Conclusions

A finite element buckling simulation analysis was performed on the afterburner cylinder of an aero-engine with a small bypass ratio, and the buckling modes and critical buckling loads were obtained. Based on simulation analysis, two structural improvement schemes were proposed. The conclusions of this study were as follows:

- Under the auxiliary mount load, buckling occurred in the front section of the cylinder corresponding to the mount load, manifesting as local buckling. Under the load of the auxiliary mounts on the left and right sides, buckling occurred in the front cylinder section corresponding to the loading of the one-side mount.
- Under the load of differential pressure (external pressure exceeding internal pressure) or the axial compressive force of the rear flange, the afterburner cylinder was highly susceptible to buckling.
- Under various rear flange loads, buckling deformation occurred near the rear flange. Under axial compressive force or torque, the buckling mode of the cylinder exhibited a regular wavy distribution. Under bending moment or lateral force, buckling occurred only in the local cylinder area near the rear flange, with the buckling deformation being closely related to the direction of the load application.
- Regarding the wall-thickening scheme, the buckling mode remained unchanged. As the wall thickness increased, the critical buckling load under all loading conditions grew linearly. An increase in the critical buckling load of 17.78% to 59.46% was observed when the wall thickness was augmented by 0.3 mm.
- For the grille reinforcement scheme, the buckling modes under the auxiliary mount load and differential pressure load were similar to the original design, but the buckling modes under the rear flange loads exhibited significant differences, compared to the original design. The critical buckling load increased by 169% to 619%, representing a substantial enhancement in load-bearing capacity.
- Although the wall thickness scheme could increase the critical buckling load and was relatively simple to implement, the increased wall thickness would result in a higher cylinder weight, negatively affecting the specific thrust-to-weight performance of the aero-engine. The grille reinforcement scheme effectively improved the rigidity and stability of the overall aero-engine while significantly increasing the critical buckling load. Therefore, the grille reinforcement scheme was deemed superior.

In current practice, chemical milling technology is commonly used for manufacturing grille-reinforced structures. This process demands high precision and has specific material requirements, leading to increased production costs and potential weight gain. It is recommended that structural optimization design be employed when utilizing a grille reinforcement scheme. Multi-objective, multi-parameter structural optimization should be conducted while considering factors such as wall thickness, stiffener layout, height, width, and fillet radius. This optimization approach aims to achieve the lightest weight while maintaining optimal strength, stiffness, and stability for aero-engine structures.

Due to the requirements of higher thrust and the thrust–weight ratio of the engine, there are a large number of thin-walled components in the aero-engine. The loads of the aero-engine are extremely complex and harsh, and most thin-walled components have the risk of buckling instability during work, which poses a potential threat to the overall safety of the aero-engine.

This paper obtains the critical buckling loads and buckling modes of the afterburner cylinder and proposes two effective anti-buckling improvement measures. This study not only contributes valuable insights into the structural behavior of afterburner cylinders in aero-engines with low bypass ratios, it also provides a foundation for the development of effective strategies for enhancing their structural stability and load-bearing capacity.

**Author Contributions:** Conceptualization, X.Z. and Y.Z.; methodology, X.Z.; software, Y.Z.; validation, B.H.; formal analysis, Y.Z. and B.H.; investigation, J.X.; resources, Z.L.; data curation, Q.Y.; writing—original draft preparation, X.Z.; writing—review and editing, Z.L.; visualization, Y.Z.; supervision, J.X. and Z.L.; project administration, X.Z. and B.H.; funding acquisition, X.Z. All authors have read and agreed to the published version of the manuscript.

**Funding:** Funding was provided by the Special Project Support for Research and Development of Key Core Technologies and Common Technologies in Shanxi Province (2020XXX017) and the Basic Research Program of Shanxi Province (Youth Science Research Project, 202203021222129).

**Institutional Review Board Statement:** Not applicable.

**Informed Consent Statement:** Not applicable.

**Data Availability Statement:** Not applicable.

**Conflicts of Interest:** The authors declare no conflict of interest.

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
