# Peer review of "Buckling Analysis and Structure Improvement for the Afterburner Cylinder of an Aero-Engine"

_aerospace, doi:10.3390/aerospace10050484_

Round 1

Reviewer 1 Report

The manuscripts deals with buckling analysis of the afterburner cylinder in military turbofan aero engines. Simulations and analysis of various loads allow obtaining buckling modes, critical buckling load, and buckling positions. Two anti-buckling improvements are suggested.

At a glance, the manuscript appears to be interesting but absolutely nothing is unarguably concluded. First of all, it is unclear whether this is a proof of concept (hence the findings can be further used in aerospace technology) or this just applies to the specific conditions of the afterburner cylinder under analysis. It is extremely vague in parts. The authors should note that the aerospace language is quantitative, thus only figures, numbers, units, and context are allowed.

The manuscript could be accepted for publication provided extensive review is carried out.

Specific comments:

1. Abstract should be re-written including values and units, and present tense please. As presented, it is just qualitative information without significance.

2. Anti-buckling improvements have already been discussed in the literature. Please do further investigation in the field and include them in the literature review and discussion. Try to highlight the novelty of this study. 

3. Justify better the use of linear buckling analysis. As such, it is vague.

4. In presenting the structure of the afterburner, it is provided its overall axial length, but not the lengths of the cylindrical and conical sections. This is a key parameter to carry out a proper buckling analysis, especially if the findings are intended to be applied into other engine designs.

5. Temperature distribution on afterburners subjected to thicker walls and/or grid reinforcements will be different. However, it is not properly considered, analysed, and discussed in the manuscript. Please do include it.

6. Discussion is extremely poor; it is actually conclusions. 

7. Clarify why and how the study provides "a foundation for the development of effective strategies" in aero-engines, especially noting that both strategies have previously been discussed in the field. 

Please note that all points above only intend to address the (poor to average) quality of the manuscript. As submitted, it is just another manuscript without significance in the vast myriad of everyday submissions. Do improve it and publish something useful that can make a difference in the field.

Author Response

Point 1: Abstract should be re-written including values and units, and present tense please. As presented, it is just qualitative information without significance.

Response 1: The abstract has been revised and improved

Abstract: The buckling failure of the afterburner cylinder is a serious safety concern for aero-engines. To tackle this issue, the buckling simulation analysis of the afterburner cylinder was carried out using finite element method (FEM) software to obtain the buckling mode and critical buckling loads. It was found that the afterburner cylinder was susceptible to buckling when subjected to differential pressure or the compressive force of the rear flange. Buckling would occur when the differential pressure reached 0.4 times atmospheric pressure or the axial compressive force on the rear flange reached 222.8 kN. Buckling was also found on the front of the cylinder under the auxiliary mount load. Additionally, under various loads on the rear flange, buckling occurred in the rear section, with the buckling mode being closely related to the load characteristics. Based on the simulation results and structural design requirements, two structural improvements were proposed, including wall thickening scheme and grid reinforcement scheme. FEM simulation analysis results showed that both schemes would improve the rigidity and stability of the afterburner cylinder, for the 0.3 mm increase in wall thickness scheme, the critical buckling load increased by 17.78 % to 59.46 %; for the grid reinforcement scheme, the critical buckling load increased by 169 % to 619 %. Therefore, the grid reinforcement scheme had stronger anti-buckling ability and was deemed the optimal solution. The findings of this paper would provide technical support for the structural design of large-size and thin-walled components of aero-engine.

Point 2: Anti-buckling improvements have already been discussed in the literature. Please do further investigation in the field and include them in the literature review and discussion. Try to highlight the novelty of this study.

Response 2:The anti-buckling has been added to the literature review and its advantages have already been highlighted and improved.

Point 3: Justify better the use of linear buckling analysis. As such, it is vague.

Response 3: For the linear buckling,add the explain in the paper:

“However, the computational cost of post-buckling analysis is higher than that of linear buckling analysis. In order to save computational costs, improve computational efficiency and ensure structural safety, linear buckling has been used widely in engineering design, which can not only evaluate critical loads quickly, but also can obtain buckling modes and guide structural anti-buckling design. To ensure the safety of components as well as save costs, this paper uses the linear buckling method for simulation analysis.”

Point 4: In presenting the structure of the afterburner, it is provided its overall axial length, but not the lengths of the cylindrical and conical sections. This is a key parameter to carry out a proper buckling analysis, especially if the findings are intended to be applied into other engine designs.

Response 4: Added axial length description.

“the axial length of the front part and second part is 333 mm and 667 mm respectively”

Point 5: Temperature distribution on afterburners subjected to thicker walls and/or grid reinforcements will be different. However, it is not properly considered, analysed, and discussed in the manuscript. Please do include it.

Response 5: Based on the small temperature gradient of the afterburner cylinder and practical engineering experience, temperature has a slight influence on wall thickness thickening or grid strengthening schemes, has a slight influence on buckling. Therefore, the improvement scheme proposed in this papaer does not consider the influence of improved temperature on the results, the temperature distibution is consistent with Figure 3.

Point 6: Discussion is extremely poor; it is actually conclusions.

Response 6: The discussion and structural improvement have been merged and improved.

Point 7: Clarify why and how the study provides "a foundation for the development of effective strategies" in aero-engines, especially noting that both strategies have previously been discussed in the field.

Response 7: The author elaborated and explained the above issues at the end of the paper

“Due to the requirements of higher thrust and thrust-weight ratio of engine, there are a large number of thin-walled components on the aero-engine. The loads of the aero-engine are extremely complex and harsh, and most thin-walled components have the risk of buckling instability during work, which posing a potential threat to the overall safety of the aero-engine.

This paper obtains the critical buckling load and buckling mode of afterburner cylinder, and proposes two effective anti-buckling improvement measures. This study not also contributes valuable insights into the structural behavior of afterburner cylinders in aero-engines with low bypass ratios, but also provides a foundation for the development of effective strategies for enhancing their structural stability and load-bearing capacity.”

Reviewer 2 Report

1. References Nr. 19 (1934) (with all my respect to ASME,  also as a member of ASME :) ) and 20 (1941), please replace with some newer one (1990-2000 may be?!)

Author Response

Point 1: References Nr. 19 (1934) (with all my respect to ASME,  also as a member of ASME :) ) and 20 (1941), please replace with some newer one (1990-2000 may be?!).

Response 1:

The theory of buckling was proposed by early scientists, author has replaced one references and retained the other one.

Reviewer 3 Report

For the paper to be accepted the authors must add a short section in the discussion and/or the conclusion to explain why their analytical exercise is a reliable study in a practical context.  It is not a validation that is needed but a detailed form of verification.  I have not checked the equations, the authors are encouraged to do so just in case.  The use of English can be improved in some places.  The legends of some figures should be made more legible.

Author Response

Point 1: For the paper to be accepted the authors must add a short section in the discussion and/or the conclusion to explain why their analytical exercise is a reliable study in a practical context.

Response 1: The supplementary content was showed at the end o section 3.3

Point 2: I have not checked the equations, the authors are encouraged to do so just in case. 

Response 2: The equations i have been verified and checked

Point 3: The use of English can be improved in some places. 

Response 3: The English expression of the paper has been improved

Reviewer 4 Report

After a detailed study of the article, it can be concluded that its topic is very current and interesting for readers. I recommend adding at least the following brief information to the text of the article:

It would be appropriate to describe in more detail how the load values in Table 2 were determined. Are the stated "round values" based on some calculation, or is it more of a "qualified guess"?

The heat transfer process in the afterburner cylinder is quite complex, especially when it comes to the local values of the heat transfer coefficients. It would therefore be appropriate to add at least a brief description of the used computational model to Figure 3, where the temperature distribution is shown.

Regarding the formal organization of the article, the basic requirements of the template are respected. However, I recommend some minor corrections:

When writing physical dimensions for numerical quantities, it is appropriate to separate this with a space, which is present in some places and not in some places (Table 2, line 197, 204, etc.).

Multiplication in mathematical relationships is not appropriate to write with the symbol * (Table 2, line 197, 204, Table 3, etc.)

Author Response

Point 1: It would be appropriate to describe in more detail how the load values in Table 2 were determined. Are the stated "round values" based on some calculation, or is it more of a "qualified guess"?

Response 1: The loads has been explained in the paper.

Point 2: The heat transfer process in the afterburner cylinder is quite complex, especially when it comes to the local values of the heat transfer coefficients. It would therefore be appropriate to add at least a brief description of the used computational model to Figure 3, where the temperature distribution is shown.

Response 2: Add description of temperature field distribution.

“According to engineering experience, the temperature distribution of afterburner cylinder gradually increases along the axial direction. ”

Point 3:When writing physical dimensions for numerical quantities, it is appropriate to separate this with a space, which is present in some places and not in some places (Table 2, line 197, 204, etc.).

Response 3: The numerical quantities and physical dimensions have been separated by spaces for the whole paper.

Point 4:Multiplication in mathematical relationships is not appropriate to write with the symbol * (Table 2, line 197, 204, Table 3, etc.)

Response 4: Author has changed * to • for the whole paper.

Round 2

Reviewer 1 Report

While the authors have tried to answers all points raised in review one, it is absolutely clear they have done it trying to please the reviewer, not necessarily to dramatically improve the quality of manuscript, which was ultimately expected.

Many of the points raised still remain: vagueness, many qualitative assessments, lack of critical comparison to existing anti-buckling approaches, etc.

Should the editorial office consider it appropriate, the manuscript might be accepted for publication. There is no need of seeing the manuscript again.

Author Response

Point 1:The Vagueness of pictures

Response 1:The pictures of whole paper have been modified and the quality has been improved

Point 2: Conduct comparative analysis of areas with data;

Response 2:Author has conducted in-depth comparative analysis of the valid data:.

Point 3:. Entry content: "Comparative analysis of optimization results and existing results"

Response 3:Author has added a comparison with the original scheme in section 4.3.